# The Prevalence of Overweight Status among Early Adolescents from Private Schools in Indonesia: Sex-Specific Patterns Determined by School Urbanization Level

**DOI:** 10.3390/nu14051001

**Published:** 2022-02-27

**Authors:** Eveline Sarintohe, Junilla K. Larsen, William J. Burk, Jacqueline M. Vink

**Affiliations:** 1Behavioural Science Institute, Radboud University, 6500 HE Nijmegen, The Netherlands; junilla.larsen@ru.nl (J.K.L.); william.burk@ru.nl (W.J.B.); Jacqueline.vink@ru.nl (J.M.V.); 2Psychology Faculty, Maranatha Christian University, Bandung 40164, West Java, Indonesia

**Keywords:** obesity, overweight, developing countries, Indonesia, adolescents, sex differences, demographic, high SEP

## Abstract

(1) Background: Few studies have investigated (demographic) correlates of (prevalent) overweight rates among early adolescents, especially from higher socioeconomic positions (SEP) in developing countries, such as Indonesia. The current study aims to fill this gap. (2) Methods: Participants included 411 adolescents from five private schools in Indonesia. Adolescents’ weight and height were measured, and adolescents completed questionnaires on demographic factors (i.e., sex, school area, ethnicity, pocket money) and previous year dieting. (3) Results: Results showed that more than one-third of the sample was overweight, with higher rates among adolescent males (47%) than females (24%). Moreover, adolescents attending schools in urban areas (compared with suburban areas), and those reporting past dieting (compared with those reporting no dieting) had higher overweight rates. Ethnicity and the amount of pocket money were not related to overweight status. Finally, a clear sex-specific interaction was found involving school area, showing that males in urban areas had a significantly higher risk to be overweight, whereas this did not apply to females. (4) Conclusions: males from urban area private schools in Indonesia may be an important target group for future preventive overweight interventions.

## 1. Introduction

The prevalence of obese and overweight individuals has continued to increase over the past years, particularly in developing countries, such as Indonesia [1,2,3]. In contrast to Western countries, obesity is positively related to socioeconomic position (SEP) in many developing countries [4,5,6], meaning that being overweight is more prevalent among adults and adolescents with higher SEP. So far, studies have suggested that in developing nations, people with a higher SEP, compared with lower SEP, have easier access to junk food or calorie-dense foods, which may explain the higher overweight rates, particularly in these groups [5,6].

Adolescence is a particularly vulnerable period for the development of overweight, not only in Western countries but also among developing countries such as Indonesia [7,8]. Moreover, overweight prevalence seems to show sex-specific differences in Indonesia. Among adolescents, the prevalence of overweight was higher in females than in males [4,9]. The same has been found for adult populations in Indonesia (i.e., higher prevalence rates among women compared with men) [2,4,10]. However, among children, the prevalence of overweight was higher in boys compared with girls [4]. These (review) results suggest some shifting sex-specific patterns during early adolescence regarding the prevalence of overweight status. The current study has a specific focus on early adolescence, a critical stage where (gender-specific) lifestyle choices change, for example, because of the availability of energy-dense (junk) food and increasing peer influences in (changing) school environments [9,11,12], with the highest possible impact among adolescents at private schools [4].

In general, school is an environment in which adolescents spend much of their time with peers, and where junk food is available. As such, school area (i.e., urban versus non-urban areas or less urban) might play an important role in obesogenic behaviors, including junk food consumption [9,13]. In line with this, some studies in Indonesia have shown that overweight patterns differ according to school area [11,14]. Adolescents living in urban (school) areas have higher prevalence rates of overweight status compared with those living in less urban or rural areas [11,14]. This might be explained by their greater access to more types of junk food or fast food compared with people living in less urban or rural areas [9,13,14,15]. Moreover, previous research has also found that overweight patterns differ according to ethnicity. To date, overweight status is more prevalent among people with Orang Asli Malaysian compared with Chinese Malaysians backgrounds [13]. Further research is needed to examine whether and how overweight status among early adolescents at private schools, with generally higher overweight risk, might differ according to school area and ethnic background in Indonesia.

Furthermore, given that the amount of pocket money may be an indication of the possibility to buy fast food, pocket money may also be related to overweight status among early adolescents from private schools and mostly higher SEP backgrounds. Finally, studies in industrialized countries (in Europe and America) have shown that dieting behavior is associated with greater weight gain over time among adolescents [16]. Due to the increasing impact of Western society on developing countries such as Indonesia, it is important to identify whether dieting behavior is similarly linked to overweight status among early adolescents at relatively high risk for being overweight from private schools in Indonesia.

Moreover, some previous studies among Indonesian populations have also shown sex-specific links regarding demographic factors explaining overweight status. Specifically, two previous Indonesian studies in adult samples showed that females from urban areas were at higher risk to be overweight compared with males [2,5]. In contrast, one recent large-scale study among children and adolescents (10–18 years old) has shown that, specifically, males living in urban areas were more likely to be overweight and obese than females living in urban areas [12]. Given these contrasting findings, it is important to further examine sex-specific links between school area and (over)weight status among early adolescents, particularly among those from private schools with higher overweight prevalence [4]. Another study suggested that overweight status may be more strongly linked to ethnicity among males compared with females [17]. Finally, some studies among Western countries also show sex-specific links between dieting and (over)weight status [14,16]. As such, we will also explore sex-specific demographic or dieting correlates of overweight status among early adolescents from private schools (higher SEP background).

To conclude, recent research on demographic correlates of overweight prevalence rates among early adolescents from private schools in Indonesia is limited. However, society, particularly around private schools (higher SEP backgrounds), might have changed rapidly with regard to eating behavior in Indonesia (more fast-food restaurants, larger influence from Western society [2,9]) making it urgent to explore correlates of overweight in early adolescents at private schools in Indonesia nowadays. As such, the aim of the current study is to examine (sex-specific) correlates of overweight status in a relatively large sample of early adolescents from private schools.

## 2. Materials and Methods

### 2.1. Participants and Procedure

The participants in this study were part of the baseline measurement (Wave 1) from an ongoing longitudinal study on adolescents’ weight-related behavior in Indonesia. Wave 1 took place in October until December 2019. Adolescents were recruited through five private junior high schools in four cities (Jakarta, Surabaya, Bandung, and Manado) in Indonesia. A total of 411 students (47.7% females) participated. All adolescents (M_age_ = 12.02 years; SD _age_ = 0.45; range = 11.02 to 14.11 years) were in 7th grade or in their first year of junior high school.

A letter describing the longitudinal project was initially sent to officials of school foundations (some private schools are organized by private foundations) or directly to school officials. If the school foundations provided approval, the agreement letter was then sent to the principal of the schools. School officials informed both the parents and students about the goals of the project. Parents were asked to return a signed consent form indicating they agreed to their child’s participation. Students were also asked to return a signed consent form indicating whether they agreed to participate in the study. Of the five schools that agreed to participate, three schools obtained consent forms from parents and students. The remaining two schools informed the parents about this project (passive consent) based on the school policy and collected the signed consent forms from students only. The original and amended (passive consent) procedures were approved by the Ethics Committee Social Science of Radboud University, Nijmegen, The Netherlands (ECSS-2019-115).

Researcher informed students that their participation was voluntary, that answers would be processed confidentially and would be stored separately from personal data (with a key file to link the data), and that they could withdraw from the study at any time. Adolescents completed a paper self-report survey at school during one classroom hour (approximately 60 min). In addition, adolescents’ weight and height measures were taken by the researcher with the assistance of the school nurse. Weight and height of participants were assessed using school equipment (stadiometer). Students were rewarded with a small gift when they completed the questionnaires.

### 2.2. Measurements

#### 2.2.1. Anthropometric Measurements

Adolescents’ height was measured to the nearest 0.1 cm with a validated stadiometer (Seca around 217), and their weight was measured to the nearest 0.1 kg with a weighing scale (Seca around 840). Based on the Center for Disease Control and Prevention (CDC) 2000 Body Mass Index for age growth charts for males and females, the cut-off for defining overweight was based on the sex and age in months and BMI (weight (kg)/height (m^2^)).

#### 2.2.2. Demographic Characteristics

Adolescent’s sex was coded, with 0 = female and 1 = male. School area was coded as 0 = suburban (Bandung and Manado) and 1 = urban area (Jakarta and Surabaya). We divided the area based on modernization and levels of Westernization [8]. There were no exclusion criteria involved. All students from urban and suburban areas participated in this study. Moreover, ethnicity was coded as 1 = Javanese, 2 = Sundanese, 3 = Sulawesi, 4 = Tionghoa (Chinese Indonesian), 5 = other ethnic (Papua, Kalimantan, Sumatra, and Bali), and 6 = mixed ethnicity. The percentage of Chinese-Indonesian ethnic students was almost half of the sample (49.8%), so we decided to dichotomize ethnicity as 0 = Indonesian ethnic (Javanese, Sundanese, Sulawesi, etc.) and 1 = Chinese Indonesian. The amount of pocket money was coded as 1 = < IDR 500,000, 2 = IDR 500,000–IDR 1,000,000, 3 = IDR 1,100,000–IDR 1,500,000, 4 = IDR 1,600,000–IDR 2,000,000, 5 = IDR 2,100,000–IDR 2,500,000, and 6 = > IDR 2,500,000. The percentage of students with pocket money less than IDR 500,000 was more than half of the sample (63%), so we decided to dichotomize pocket money as 0 = < IDR 500,000 and 1 = ≥ IDR 500,000.

#### 2.2.3. Dieting Behavior

To measure previous dieting behavior, participants were asked, “In the past year, how often did you diet in an attempt to have the same weight or lose weight?” The response categories for this item were: 1 = never, 2 = 1–2 times, 3 = 3–4 times, 4 = 5–6 times, and 5 = 7 times or more often. Initial inspection of the distribution of this item indicated a substantial group of adolescents who reported no past year dieting (50.68%), so this item was also dichotomized as 0 = no past year diet and 1 = did past year diet.

### 2.3. Statistical Analyses

Chi-square analyses were performed to examine univariate demographic and dieting differences between overweight and non-overweight groups. Moreover, a logistic regression analysis was performed to explain overweight status group membership (0 = not overweight; 1 = overweight) from several predictors. The independent variables included in this analysis were student´s sex, school area (suburban vs. urban), ethnicity (Indonesian vs. Chinese Indonesian), pocket money (<500,000 vs. ≥500,000), and previous dieting behavior (never vs. did diet). Moreover, sex-specific interactions (i.e., sex by school area, sex by ethnicity, sex by pocket money, and sex by previous diet) were tested in four separate analyses (one interaction per analyses added to the main effects model). Statistically significant interactions were further probed using the PROCESS module in SPSS [18].

## 3. Results

### 3.1. Descriptive Statistics

Data from a total of 411 students were examined in this study. The sample was equally divided according to sex (53.3% boys). In total, 59.1% of the adolescents attended a school located in an urban area. The sample was also equally divided according to ethnic background (51.2% Indonesians and 48.8% Chinese Indonesians). Most of the respondents had less than IDR 500,000 per month (63%) and had not dieted (50.6%).

In the total sample, 36.3% of the adolescents were characterized as being overweight. Chi-square independence tests indicated that overweight status was more prevalent in males compared with females (see Table 1). Moreover, adolescents living in urban school areas had a higher overweight prevalence compared with those living in suburban areas. Finally, the adolescents reporting previous dieting were more likely to be overweight compared with those who did not report dieting. Overweight status did not differ according to ethnic background or amount of pocket money.

### 3.2. Unique Contributions of Demographics and Dieting in Explaining Overweight Status

A binary logistic regression was performed to examine the unique contributions of the five predictors in explaining overweight status. Males were 3.28 times more likely to be overweight compared with females (CI 95% (2.08, 5.18)). Adolescents from urban school areas were 1.84 times more likely to be overweight compared with those from suburban school areas (CI 95% (1.07, 3.14)). Adolescents reporting dieting were 3.84 times more likely to be overweight compared with their non-dieting counterparts (CI 95% (2.45, 6.03)). The main effects of ethnicity and pocket money were not statistically significant (see Table 2). All variables together explained 20.3% of the variance in overweight status.

### 3.3. Sex-Specific Interactions

Four separate sex-specific interaction analyses were performed, in which one interaction was added to the main regression model. Of these four interactions, only the interaction between sex and school area was statistically significant (see Table 3). The explained variance for the total model including the interaction was 21.5% (b = 0.99, SE = 0.47, CI 95% (0.07, 1.89)). We further probed this interaction using Model 1 PROCESS module for SPSS. The results showed that males living in urban areas were more likely to be overweight compared with males living in suburban areas (b = 0.99, SE = 0.48, and CI 95% (0.34, 1.65)), whereas this did not apply to females (b = 0.01, SE = 0.39, and CI 95% (−0.75, 0.78)). The other sex-specific interactions were not statistically significant.

## 4. Discussion

The current study aimed to examine the (sex-specific) demographic and dieting factors that potentially explain overweight status among a relatively large group of Indonesian early adolescents attending private schools (i.e., higher SEP background). Children and adolescents from private schools are more likely to be obese [4] and early adolescents’ weight is predictive of their weight status in adolescence and adulthood [11]. Finding correlates of overweight in this specific period might give insights for future prevention or intervention and may have both direct and longer-term health benefits. Our findings showed that the general prevalence rate of overweight in this early adolescent sample at private schools was relatively high (i.e., 36.3%) compared with previous national prevalence rates (i.e., 16%) [11]. The seemingly higher percentage of overweight status among adolescents from higher SEP backgrounds may be environmentally driven [4]. As mentioned, higher SEP private schools, particularly those in urban environments, are often located in areas with more junk food outlets [3,9]. Food outlets usually sell fried products, that are highly preferred, and these kinds of products are highly energy dense. People from higher SEP backgrounds often opt to eat out rather than at home, and food served in restaurants or food outlets usually contains more calories [3,15,19]. In addition, most Indonesian parents from higher SEP are proud when their children look big or fat, reflecting a higher socioeconomic status [3]. Together, these factors may probably explain the relatively high overweight prevalence rate in our study sample.

The relatively high overweight prevalence in our sample makes further insights into (sex-specific) demographic correlates even more interesting, given the increased statistical power to detect effects. We found that males were almost four times more likely to be overweight than females. This result is consistent with most previous studies among Indonesian children [4,9]. However, these findings are in contrast with previous studies among adolescents and adults, where prevalence rates are mostly reported to be higher among females compared with males [2,4,10]. Our findings indicate that early adolescent males are (still) more likely to be overweight compared with early adolescent females, at least among adolescents attending private schools. Future longer-term studies following early adolescents to emerging adulthood are needed to further shed light on a potential sex-specific switch in terms of overweight vulnerability. Specifically, sex-specific parental perceptions of ideal body weight among children and early adolescents may explain the higher prevalence rate of overweight status among (early adolescent) males. Parents seem more supportive of higher body weights of males compared with females, possibly because of the male body ideal (big is more ideal for males than females, [20,21]). As such, these explanations may thus explain our sex-specific findings involving overweight status, given that parents may still have a considerable influence on what their children eat (potentially impacting their weight development) during early adolescence [22].

We also found a significantly higher prevalence of overweight status among Indonesian adolescents who attended schools in urbanized areas compared with those in suburban areas. This has similarly been reported before among children and adolescents [4,11,23]. However, this finding should be interpreted carefully in our case because we also found a clear sex-specific interaction with school area. We found that specifically male adolescents in urban school areas had higher overweight rates. This finding is in line with another recent study among children and adolescents in Indonesia [11], but in contrast with previous studies among adults showing that females from urban areas were the ones at highest risk [2,5]. We speculate that (early adolescent) males may be more vulnerable to these unhealthy urban environments with junk food cues from fast-food outlets, as they often show higher efforts to get food as a reward compared with females [22,24]. As such, males might be more likely to actively search for food rewards, which are more often satisfied in high junk food environments. This, in combination with parents possibly more often encouraging adolescent males to gain weight [21], might explain our sex-specific interaction among early adolescents.

In our study, ethnicity was not related to overweight status, which is in contrast with some previous studies [13,25]. However, our findings involving ethnic background are consistent with the results of a previous study investigating adolescents from other Indonesia regions (i.e., Surakarta). This study also found no significant differences between Javanese and Chinese Indonesian adolescents [17]. So it might be that ethnicity findings regarding overweight status are dependent upon the specific Indonesian region (and ethnicities) being examined.

A final result of our study is that adolescents who dieted in the previous year were more likely to be overweight. This finding is in line with well-known findings from Western countries, with recent dieting considered to be a potential proxy of the susceptibility to weight gain [26]. It might be that dieting is unsuccessful and interspersed with binge eating episodes, thus leading to weight gain. Dieting may also be the consequence of being overweight [16]. Further longitudinal research is needed to unravel the directionality of these associations. Importantly, the fact that the dieting findings in this study were rather similar to the ones reported in previous European and American studies, suggests overlap in terms of overweight correlates between higher SEP Indonesian adolescents and adolescents from Western countries.

One notable strength of our study is the inclusion of a relatively large sample of early adolescents from specifically private schools, who are at higher risk for obesity, as also supported by our study findings. Another strength is that we used objectively measured weight and height to determine overweight status. Nevertheless, a couple of limitations should also be mentioned. First, we did not include clear markers for determining “socioeconomic” differences (except pocket money) within our higher SEP group of adolescents from private schools. The amount of pocket money that adolescents received might not reflect socioeconomic position differences. The income of the family per year might have been a better indicator (i.e., [5,12]). Nevertheless, as our total sample was recruited from private schools only, we are rather confident that most adolescents were from mid-to-high SEP backgrounds. Second, as our data are limited by a cross sectional design, we, therefore, do not know the underlying mechanism explaining the observed associations. Future longitudinal studies could shed more light on (predictors of) weight development in specific subgroups, such as males from urban areas compared with suburban areas.

Despite these limitations, our study examining (sex-specific) demographic correlates of overweight status among early adolescents from private schools in Indonesia filled an important gap in the current literature. We have speculated about the most prominent (mostly nutrient-related) mechanisms explaining our findings. Nevertheless, future research should further unravel the underlying (energy intake and expenditure) mechanisms explaining why particularly early adolescent males from urban school areas are more likely to be overweight. This will provide further tools for future tailored preventive interventions. We suggest that this early adolescent phase is a promising period for timely preventive interventions, given that adolescent overweight and obese status in Indonesia is more rapidly increasing in older compared with younger adolescents [8]. To conclude, our findings suggest that males from urban area private schools in Indonesia may be an important target group for future preventive overweight interventions.

## Figures and Tables

**Table 1 nutrients-14-01001-t001:** Chi-square analyses examining adolescent’s overweight status differences as a function of demographic and dieting characteristics.

	Not Overweight		Overweight		Chi Square	*p* Values
	*n*	%	*n*	%		
**Sex**						
Females	146	46	24.0	24.0	23.57	<0.001
Males	116	103	47.0	47.0		
**Ethnicity**						
Indonesian (Ethnicities)	135	65.5	71	34.5	0.63	0.428
Chinese Indonesian	126	61.8	78	38.2		
**School Area**						
Suburban	117	69.6	51	30.4	4.27	0.039
Urban	145	59.7	98	40.3		
**Pocket Money**						
<500,000	166	64.1	93	35.9	0.04	0.849
≥500,000	96	63.2	56	36.8		
**Diet**						
Never diet	160	76.9	48	23.1	31.64	<0.001
Diet (1—more than 5 times)	102	50.2	101	49.8		

**Table 2 nutrients-14-01001-t002:** Logistic regression predicting overweight status by demographic and dieting correlates in the total group.

	B	SE	OR	CI 95%	Nagelkerke R^2^
Sex	1.19 **	0.23	3.28 **	2.08–5.18	
School area	0.61 *	0.27	1.84 *	1.07–3.14	
Ethnicity	−0.11	0.26	0.90	0.54–1.51	20.3
Pocket money	0.02	0.23	1.02	0.64–1.61	
Past year dieting	1.35 **	0.23	3.84 **	2.45–6.03	

Note: * *p* ≤ 0.05, ** *p* ≤ 0.01. Sex: 0 = females, 1 = males; school area: 0 = suburban, 1 = urban; ethnicity: 0 = Indonesian, 1 = Chinese Indonesian; pocket money: 0 = <500,000 Rp, 1 = >500,000 Rp; and past year dieting: 0 = no dieting, 1 = dieting. B: Beta; SE: Standard Error; OR: Odds Ratio; CI: Confidence Interval.

**Table 3 nutrients-14-01001-t003:** Logistic regression predicting overweight status by demographic and dieting correlates in the total group including interaction effects with sex.

	B	SE	OR	CI 95%	Nagelkerke R^2^
Sex	1.19 **	0.23	3.28 **	2.08–5.18	
School area	0.61 *	0.27	1.84 *	1.07–3.14	
Ethnicity	−0.11	0.26	0.90	0.54–1.51	20.3
Pocket money	0.02	0.23	1.02	0.64–1.61	
Past year dieting	1.35 **	0.23	3.84 **	2.45–6.03	
Sex *school area	0.98 *	0.47	2.67	1.07–6.63	21.5
Sex * pocket money	−0.61	0.47	0.54	0.21–1.38	20.7
Sex * ethnicity	0.87	0.46	2.39	0.97–5.90	21.3
Sex * past year dieting	−0.46	0.49	0.63	0.24–1.66	20.5

Note: * *p* ≤ 0.05, ** *p* ≤ 0.01. Two-way interactions were tested separately (one interaction per analyses added to the main model). Total model explained variance were reported per separately tested interaction. Sex: 0 = females, 1 = males; school area: 0 = suburban, 1 = urban; ethnicity: 0 = Indonesian ethnic, 1 = Chinese-Indonesian ethnic; pocket money: 0 = <500,000, 1 = >500,000; and past year dieting: 0 = never did diet, 1 = did diet. Sex-specific interactions (Sex X school area): girls = b = 0.01 (CI95% (−0.75, −0.78)); boys = b = 0.99 (CI95% (0.34–1.65)) using Model 1 PROCESS module for SPSS.

## Data Availability

The datasets generated and analyzed during the current study are not publicly available due to agreements we have made concerning the exchange and use of our data but are available from the corresponding author (E.S.) on reasonable request.

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
