# Peer review of "The Prevalence of Overweight Status among Early Adolescents from Private Schools in Indonesia: Sex-Specific Patterns Determined by School Urbanization Level"

_nutrients, 2022, doi:10.3390/nu14051001_

Round 1
Reviewer 1 Report
Sarintohe, et al. present a manuscript to analyze the factors related to increased socioeconomic status and obesity in developing geographical regions. They established a thought out and detailed method to examine this and carried out their statistics rigorously.
I only really have a minor issue when discussing risk throughout the manuscript. I thought it might be useful to either do a risk analysis that estimates the probability of an outcome given all possible outcomes, or change the wording within the manuscript that mentions "risk" to something else.
I thought it was a really straightforward manuscript and should be a good addition to the field.
Author Response
Sarintohe, et al. present a manuscript to analyze the factors related to increased socioeconomic status and obesity in developing geographical regions. They established a thought out and detailed method to examine this and carried out their statistics rigorously.
We would like to thank the reviewer for his or her positive comment on our methodology and statistical analyses.
I only really have a minor issue when discussing risk throughout the manuscript. I thought it might be useful to either do a risk analysis that estimates the probability of an outcome given all possible outcomes, or change the wording within the manuscript that mentions "risk" to something else.
We thank the reviewer for his or her thoughtful feedback. We agree that we cannot use the term “risk” given the analyses performed. As such, we have changed the term “risk” to other wordings throughout our manuscript. Please see: page 4, lines: 172-175; Page 6, line 239-240.
I thought it was a really straightforward manuscript and should be a good addition to the field.
We thank the reviewer for the positive words on our manuscript.
Remark of editor
English language and style are fine/minor spell check required
Our native English co-author (William Burk) has checked the spelling and minor spelling mistakes have been revised.

Reviewer 2 Report
The paper entitled “The Prevalence of Overweight among Early Adolescents from Private Schools in Indonesia: Sex-specific Patterns Determined by School Urbanization Level” presents interesting and valuable information based on observations and questionnaires of selected group of adolescents. Taking into account that the issue of obesity and overweight is very important, these type of analysis is significant for whole society.
On the whole, the paper is well prepared. Introduction provides sufficient information about the presented subject of study. Methodology is adequate. Results were presented in form of tables. Very important is statistical analysis which is provided. I have some small suggestions:
- Please provide conclusions which will be very useful for the readers
- I would enriched Discussion with analysis of dietary habits of adolescents in Indonesia because people of Europe or America do not know these habits
- Please adjust references style to the Journal guidelines
Author Response
The paper entitled “The Prevalence of Overweight among Early Adolescents from Private Schools in Indonesia: Sex-specific Patterns Determined by School Urbanization Level” presents interesting and valuable information based on observations and questionnaires of selected group of adolescents. Taking into account that the issue of obesity and overweight is very important, these type of analysis is significant for whole society.
We would like to thank the reviewer for his or her positive words on the relevance of our manuscript research.
On the whole, the paper is well prepared. Introduction provides sufficient information about the presented subject of study. Methodology is adequate. Results were presented in form of tables. Very important is statistical analysis which is provided. I have some small suggestions:
- Please provide conclusions which will be very useful for the readers
We thank the reviewer of his or her positive remarks about the paper and the analyses performed and for the suggestion. We have added a more concrete conclusion regarding the relevance of our findings for future preventive interventions. The following was added on page 7 lines 306-308:
“To conclude, our findings suggest that boys at Indonesian private schools in urban area may be an important target group for future preventive overweight interventions.”
- I would enriched Discussion with analysis of dietary habits of adolescents in Indonesia because people of Europe or America do not know these habits
We have enriched both the Discussion section with an example of the kind of energy-dense products that adolescents often consume. See the change on page 6, lines 226-227. Moreover, in response to reviewer 3 we have decided to change the term energy-dense food to junk food consistently throughout our manuscript.
- Please adjust references style to the Journal guidelines.
We indeed did not follow references style of Nutrients Journal guideline. We have revised the references following the Journal guidelines.
Remark of editor
Moreover, parents have also been reported to be more lenient towards their sons compared to their daughters in terms of fast-food products [21]. Error! Reference source not found.
We checked and this sentence is not explained in reference [21], so we deleted this sentence.

Reviewer 3 Report
Eveline Sarintohe et al in their manuscript titled ‘The Prevalence of Overweight among Early Adolescents from Private Schools in Indonesia: Sex-specific Patterns Determined by School Urbanization Level’ very elegantly discussed the prevalence of overweight school-going adolescents in different locations within the country. In my opinion, this is an interesting study. The hypothesis is clear and it’s a well-written manuscript. To improve the quality of the manuscript, I have only a few minor suggestions, my comments are below.
1. In my opinion, the Introduction section is highly repetitive (at several locations) with the word ‘energy-dense food (Junk)’ and it is better to clarify with a list of those energy-dense food products and that may avoid repetitive wording without the same context.
2. Methods section, page 3, line 133, replace the word dummy coded to blinded. Also, please provide details if there were any exclusion criteria involved with the student demographics.
3. In the discussion section, please expand and separate the study limitations (not explained in detail) and then include the strength of the study.
Author Response
Eveline Sarintohe et al in their manuscript titled ‘The Prevalence of Overweight among Early Adolescents from Private Schools in Indonesia: Sex-specific Patterns Determined by School Urbanization Level’ very elegantly discussed the prevalence of overweight school-going adolescents in different locations within the country. In my opinion, this is an interesting study. The hypothesis is clear and it’s a well-written manuscript. To improve the quality of the manuscript, I have only a few minor suggestions, my comments are below.
- In my opinion, the Introduction section is highly repetitive (at several locations) with the word ‘energy-dense food (Junk)’ and it is better to clarify with a list of those energy-dense food products and that may avoid repetitive wording without the same context.
We would like to thank the reviewer for the positive complement of our manuscript.
We have changed consistently the term energy-dense food´ to ´junk food`. We put yellow mark to the word, we have changed.
- Methods section, page 3, line 133, replace the word dummy coded to blinded. Also, please provide details if there were any exclusion criteria involved with the student demographics
We changed the word `dummy coded´ to ´coded´, as ´blinded´ is not the right word in this context. We did not perform any manipulations. Moreover, there were no exclusion criteria involved. All students at the participating private schools in urban and suburban areas were invited to participate.
- In the discussion section, please expand and separate the study limitations (not explained in detail) and then include the strength of the study.
We thank the reviewer for his or her thoughtful comment. We have expanded and separated the study limitations from the strengths of the study. This part now read as follows:
“Second, our data are limited by a cross-sectional design, we therefore do not know the underlying mechanism explaining the observed associations. Future longitudinal studies could shed more light on (predictors of) weight development in specific subgroups, like males from urban areas compared to suburban areas.” (Page 7, lines 293-296)
Remark of editor
English language and style are fine/minor spell check required
Our native English co-author (William Burk) has checked the spelling and minor spelling mistakes have been revised.
